# SARS-CoV-2 reinfections during the first three major COVID-19 waves in Bulgaria

**Georgi K. Marinov**[1]*, **Mladen Mladenov**[2], **Antoni Rangachev**[3,4], **Ivailo Alexiev**[5]

**1** Department of Genetics, Stanford University, Stanford, CA, United States of America, **2** Premier Research, Morrisville, NC, United States of America, **3** Institute of Mathematics and Informatics, Bulgarian Academy of Sciences, Sofia, Bulgaria, **4** International Center for Mathematical Sciences-Sofia, Sofia, Bulgaria, **5** National Center of Infectious and Parasitic Diseases, Sofia, Bulgaria

* GKM359@gmail.com

## Abstract

### Background

The COVID-19 pandemic has had a devastating impact on the world over the past two years (2020-2021). One of the key questions about its future trajectory is the protection from subsequent infections and disease conferred by a previous infection, as the SARS-CoV-2 virus belongs to the coronaviruses, a group of viruses the members of which are known for their ability to reinfect convalescent individuals. Bulgaria, with high rates of previous infections combined with low vaccination rates and an elderly population, presents a somewhat unique context to study this question.

### Methods

We use detailed governmental data on registered COVID-19 cases to evaluate the incidence and outcomes of COVID-19 reinfections in Bulgaria in the period between March 2020 and early December 2021.

### Results

For the period analyzed, a total of 4,106 cases of individuals infected more than once were observed, including 31 cases of three infections and one of four infections. The number of reinfections increased dramatically during the Delta variant-driven wave of the pandemic towards the end of 2021. We observe a moderate reduction of severe outcomes (hospitalization and death) in reinfections relative to primary infections, and a more substantial reduction of severe outcomes in breakthrough infections in vaccinated individuals.

### Conclusions

In the available datasets from Bulgaria, prior infection appears to provide some protection from severe outcomes, but to a lower degree than the reduction in severity of breakthrough infections in the vaccinated compared to primary infections in the unvaccinated.

**Data Availability Statement:** All datasets and associated code can be found at https://github.com/Mlad-en/Cov-Reinfections.

**Funding:** The author(s) received no specific funding for this work.

**Competing interests:** The authors have declared that no competing interests exist.

## Introduction

The COVID-19 [1–3] pandemic has become the most significant public health crisis in more than a century, and is still rapidly developing. An important question for its future trajectory, especially given the large and steadily growing number of infected individuals in most countries, is the degree of protection from subsequent infection and serious disease that prior SARS-CoV-2 infection and recovery confers.

SARS-CoV-2 belongs to the coronavirus family, of which four different endemic human viruses were known prior to the pandemic—HCoV-OC43 [4, 5], HCoV-229E [6], HCoV-NL63 [7, 8] and HCoV-HKU1 [9–11]. These usually cause common colds (around 10-15% of colds, depending on the source [12–15], are considered to be caused by them), and, as is common with respiratory viruses [16], they cause repeated reinfections throughout people's lifetimes [17]. Large coronavirus epidemics are thought to occur at two- to three-year intervals [18, 19], though these are generally not noticed by society due to the overall mild nature of these viruses.

Given that SARS-CoV-2 belongs to the same family of viruses, it is natural to expect that a similar host-pathogen dynamics involving frequent reinfections will be observed with it too.

The first reports of repeated infections appeared very early in the pandemic [20]. However, at the time it was difficult to exclude the possibility of simple persistence of viral RNA as opposed to true reinfections. Viral genomic sequencing (showing that distinct viral lineages infected the same individual more than once) eventually proved beyond reasonable doubt that reinfection occurs, but it was still initially seen as an exotic and surprising phenomenon [21–27]. Since then, however, reinfection has been proven to be far from a rare phenomenon as a large body of case reports has accumulated from around the world [28–121], most recently including even cases of third infections [38, 122, 123].

A number of cohort studies have also been published [123–173], but most of these suffer from various drawbacks, such as the inclusion of a very narrow time window after initial infection, focus on healthcare workers (meaning that the age distribution is not representative of the overall population), and the fact that most such studies were carried out prior to the appearance of the more highly derived SARS-CoV-2 variants that have come to dominate the pandemic in 2021 and 2022. The importance of comprehensive population sampling was shown by a recent reinfection study from Denmark [127], which found protection from reinfection of only 47.1% among those 65 years old and older during the late-2020 surge as opposed to 80.5% for the general population. The importance of variants was first stressed by the placebo arm of the clinical trial of the Novavax vaccine in South Africa [174], which showed little protection of prior infection against infection with the dominant at the time there B.1.351 variant [175].

Later, towards the end of 2021, the Omicron lineage of variants emerged, with very strong immune escape characteristics [176–179] and the ability to reinfect convalescent individuals at a high rate [123, 180–183].

In this work, we analyze available reinfection data in Bulgaria prior to the emergence of the Omicron variant, when largely homologous antigenically variants were circulating. Bulgaria has been one of the most seriously affected by the pandemic countries [184], having experienced three major COVID-19 waves in 2020-2021 and exhibiting excess mortality approaching 1% of its population within that period [185]. In the same time, only a small portion of the population has been fully vaccinated ($\leq$30% by the end of 2021), meaning that the country provides a unique context in which the clinical impact of reinfections can be observed in a previously severely impacted population with an age structure skewed towards the elderly individuals, but without the confounding factor of high vaccination coverage. We identify 4,106

reinfected individuals out of ≤700,000 cases in the country prior to December 2021. The frequency of reinfection increased substantially during the third wave driven by the Delta variant, at which point reinfections represented ∼2.2% of cases, with protection conferred by previous infection ∼81%. The severity of reinfections (i.e. the rate of hospitalizations and fatalities) was comparable to that of primary infections, while severity was reduced in breakthrough infections in vaccinated uninfected subjects.

## Methodology

The research described in this manuscript has been approved by the Ethics Committee of the IMI-BAS (Institute of Mathematics and Informatics, Bulgarian Academy of Sciences).

### Datasets

**Primary SARS-CoV-2 Infections in Bulgaria.** At the time of writing this manuscript, there were no publicly available age-stratified datasets on hospitalizations and deaths associated with confirmed SARS-CoV-2 infections in Bulgaria. We obtained a patient-sensitive dataset from Bulgaria's Ministry of Health, which included data on all infections from the beginning of the pandemic until November 5th 2021.

This dataset included information about a person's age, gender, region, the date of their latest Covid-19 test, their status (infected, recovered, hospitalized, deceased), their hospitalization start and end dates, if any, information about accompanying diseases, as well as whether they received any breathing assistance, whether they were taken into intensive care and whether they died of Covid-19.

**Data on breakthrough infections in vaccinated individuals.** Information about infections, hospitalizations and deaths among the vaccinated population in Bulgaria were obtained through publicly available datasets provided by Bulgaria's Ministry of Health. These datasets present a daily time series that contain information about the age at 10 year intervals, gender, vaccination course and count of infected, hospitalized or deceased per group.

**Reinfections.** No publicly available datasets about the reinfection rates in Bulgaria existed prior to the writing of this manuscript. We obtained these datasets through a separate request for information on patient-sensitive data from Bulgaria's Ministry of Health. The data provided by the Ministry covers the period from the beginning of the pandemic until December 9th 2021.

Reinfections were defined as cases of two positive tests spaced ≥90 days apart.

Breakthrough reinfections were defined as cases of a second positive tests at least one day after the completion of the vaccination course.

**SARS-CoV-2 sequencing data.** Information about sequenced SARS-CoV-2 genomes was obtained from the GISAID database [186].

## Results

### Suspected SARS-CoV-2 reinfection cases in Bulgaria

In order to identify SARS-CoV-2 reinfection cases in Bulgaria, we obtained datasets on the incidence and clinical outcomes of suspected reinfections up to December 9th 2021. We classified cases as suspected reinfections if ≥90 days have passed between testing positive on at least two different occasions.

After largely successfully escaping the first global wave of infections in the first half of 2020, Bulgaria experienced three major waves of COVID-19, in October-December 2020, in February-April 2021, and in the later months of 2021, of roughly equal magnitude (Fig 1A). Under

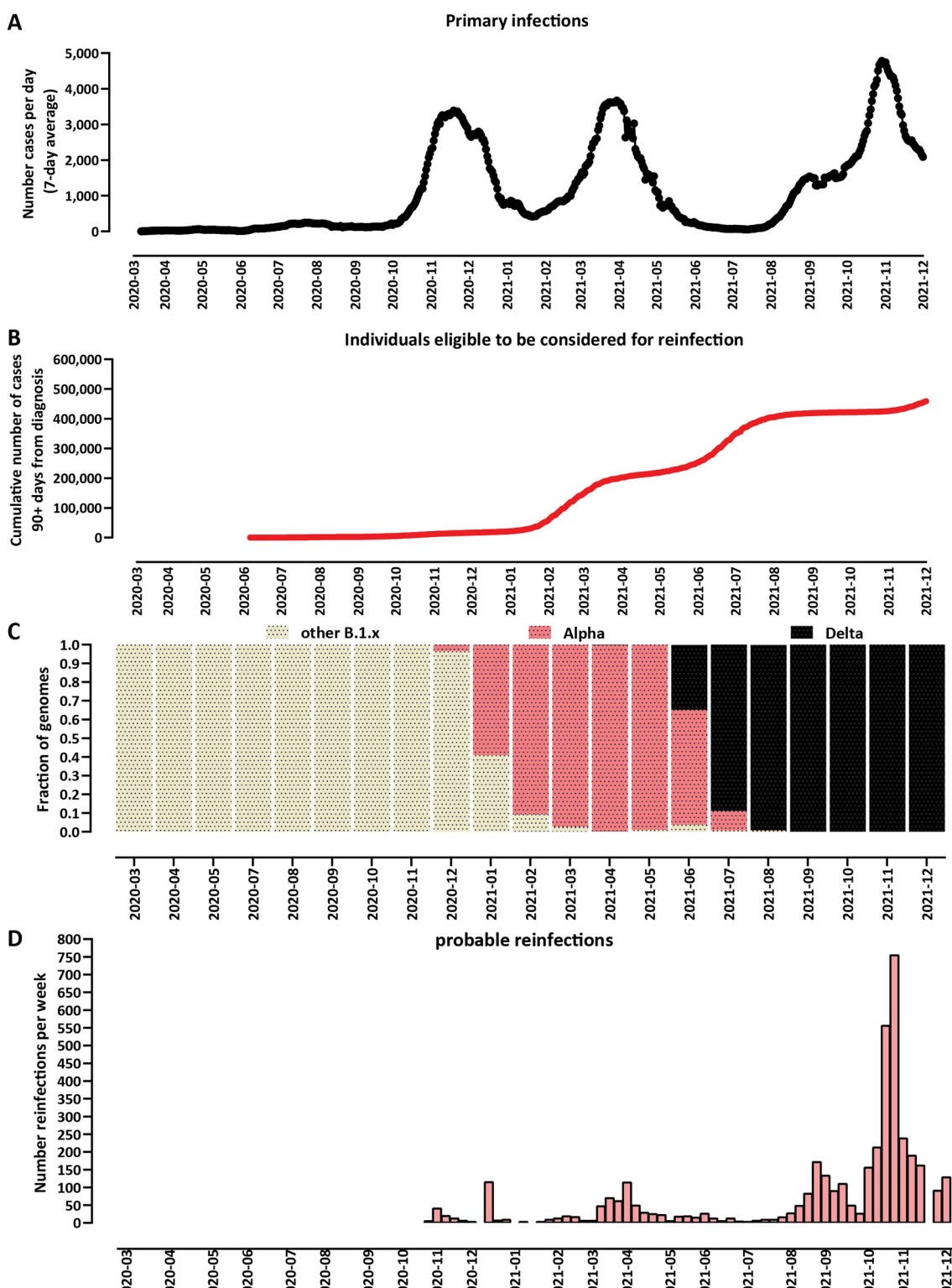

**Fig 1. Suspected SARS-CoV-2 reinfections in Bulgaria over time.** (A) Primary infections in Bulgaria over time. Bulgaria has so far experienced three distinct major epidemiological waves of COVID-19, with peaks in November 2020, March 2021, and October 2021 (an initial wave did occur in the first half of 2020 but it was very small and generally successfully suppressed, and is of little relevance to the progression of the pandemic in the country). (B) Number of people eligible to be considered for reinfection, i.e. people who have tested positive and ≥90 days have elapsed since that positive test. (C) Dominant variants in Bulgaria over time. The first major

wave was driven by early B.1/B.1.* derivative variants. The second wave was associated with the Alpha/B.1.1.7 variant. The third wave was dominated by the Delta/B.1.617.2 variant and its AY.* sublineages. (D) Number of probable reinfections over time in Bulgaria (per week).

this criterion, the eligible population to be considered for potential reinfection was $\sim$ 200,000 individuals after the first major wave, doubling to $\geq$400,000 after the second (Fig 1B). These waves were driven by different variants of the SARS-CoV-2 virus. The first was dominated by B.1 lineages antigenically similar to the ancestral strain. The second consisted almost entirely of the Alpha (B.1.1.7) variant [187, 188], while in the third the globally dominant by then Delta (B.1.6.17.2/AY.*) variant [189] constituted practically all cases (Fig 1C). We have defined for the purposes of our analyses the dividing lines between these waves as mid-January 2021 and beginning of June 2021.

In total, we identified 4,106 cases of individuals infected more than once, including 31 cases of people infected three times and one case of a quadruple infection.

The number of reinfections in the first major wave in late 2020 was small, peaking at $\leq$100 such cases weekly, reflecting the low incidence of COVID-19 earlier that year (Fig 1C). A larger, though still relatively small number of reinfections were observed during the Alpha wave in the first half of 2021. The bulk of reinfections came during the Delta wave in the second half of the year, peaking at 755 a week at the end of October 2021. During the Delta wave reinfections constituted $\sim$2.3% of cases in Bulgaria. Taking into account the number of eligible for reinfection individuals, during the months of October and November protection from reinfection is estimated to have stood at $\sim$81% (95% CI [190], 63% to 100%).

We then examined the time between primary and subsequent infections. We observe a peak at approximately a year from the initial infection, but the distribution is highly dispersed and a large number of reinfections are observed all throughout the interval from 90 to 360 days (Fig 2A). These numbers correspond primarily to a cohort of people who were infected in the first wave and then reinfected in the Delta wave ($n = 1,674$), and another group of people infected during the Alpha wave and then reinfected during the Delta wave ($n = 1,435$).

## Clinical severity of reinfections

We then analyzed the clinical outcomes of reinfections and compared it to outcomes from primary infections and from infections in vaccinated individuals ("breakthrough infections").

Among the 4,106 reinfections, 413 were also "breakthrough reinfections", i.e. the reinfection occurred after a vaccination course was completed. We divided the reinfection cases into separate unvaccinated and breakthrough reinfection categories.

A total of 84 fatalities were recorded within the reinfected cases, one of them within the set of 31 individuals with three infections. This corresponds to an apparent lower case fatality rate (CFR) than the total CFR in Bulgaria for the studied period ($\sim$2% compared to $\sim$4.2%). In terms of hospitalizations, for the 4,106 reinfected individuals, 705 hospitalizations were recorded for the second infections (a rate of 17.7%); this compares to 8,177 hospitalizations out of 49,170 breakthrough cases in vaccinated individuals (16.6%) and 109,108 hospitalizations out of 332,510 total primary infections (32.8%). However, such comparisons based on total numbers are confounded by the fact that populations are not age matched.

We therefore divided cases in all four categories into age groups and compared the rates of hospitalizations and fatalities in each (Fig 3). This analysis reveals a moderately reduced rate of hospitalizations between primary infections and reinfections across all age groups (we observe 40% reduction of risk in the 20-60 age group and 31% in the 60+ age group for the unvaccinated reinfected, and 60% and 40% for the vaccinated reinfected, respectively), and a less

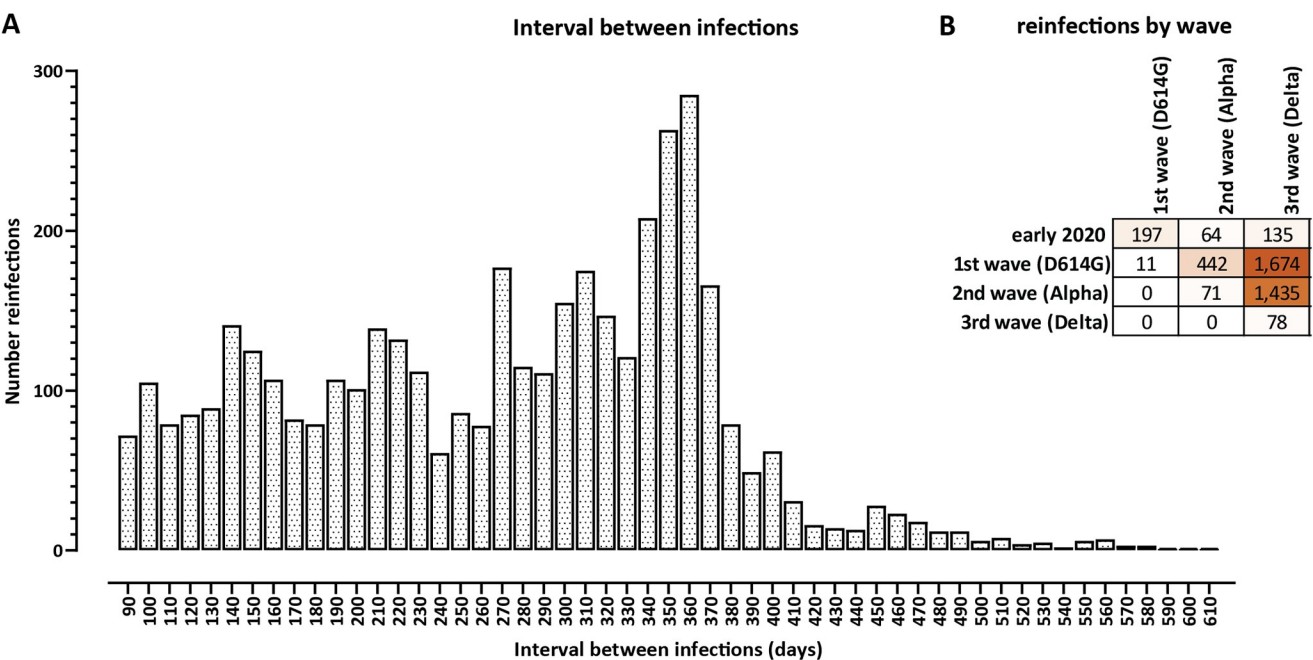

**Fig 2. Time between primary infections and reinfections against the background of different SARS-CoV-2 variants.** (A) Distribution of the length of the interval between primary infection and reinfection. (B) Primary infections and reinfections by wave and dominant variant. Waves were defined as follows: "initial infections" refers to the period prior to September 2020; "1st wave" refers to the period between September 2020 and the middle of January 2020, during which D614G variants without many other notable mutations were dominant; the "2nd wave", between mid-January 2021 and June 2021 was dominated by the B.1.1.7/Alpha variant; the "3rd wave", dominated by the B.1.617.2/Delta variant, began in July 2021.

pronounced risk of death (38% reduction in the 20-60 age group and 25% in the 60+ age group for the unvaccinated reinfected; note that total numbers were too small for break-through reinfections for an accurate estimate). For comparison the severity of breakthrough infections was more strongly reduced compared to primary infections in the unvaccinated (60%/70% risk reductions for hospitalization/death in the 20-60 age group and 49%/66% in the 60+), although that effect diminished in the higher age groups (consistent with previous findings of lower vaccine efficacy in the elderly [191–193]).

## Conclusions

In this study we evaluated the rate of incidence and the clinical outcomes of SARS-CoV-2 reinfections during the first three waves of the COVID-19 pandemic in Bugaria, and compared them to primary infections and breakthrough infections in vaccinated individuals. The bulk of reinfections happened during the Delta variant-driven wave, with prior infection providing protection from reinfection at ∼80%. Clinical severity was somewhat reduced relative to primary infections, but to a lesser extent than the observed reduction in severity in breakthrough infections in the vaccinated. A possible limitation of our study is the possibility that in some individuals the disease may have passed with mild symptoms or asymptomatically, and thus not all cases have been properly diagnosed and registered in the national system, leading to some bias towards documenting symptomatic infections. Results regarding the relative severity of reinfections in the literature have ranged from finding no difference in the severity of reinfections and primary infection to finding considerable (though rarely very high) degree of reduction from severe outcomes [169]; our results also fit within this range of estimates.

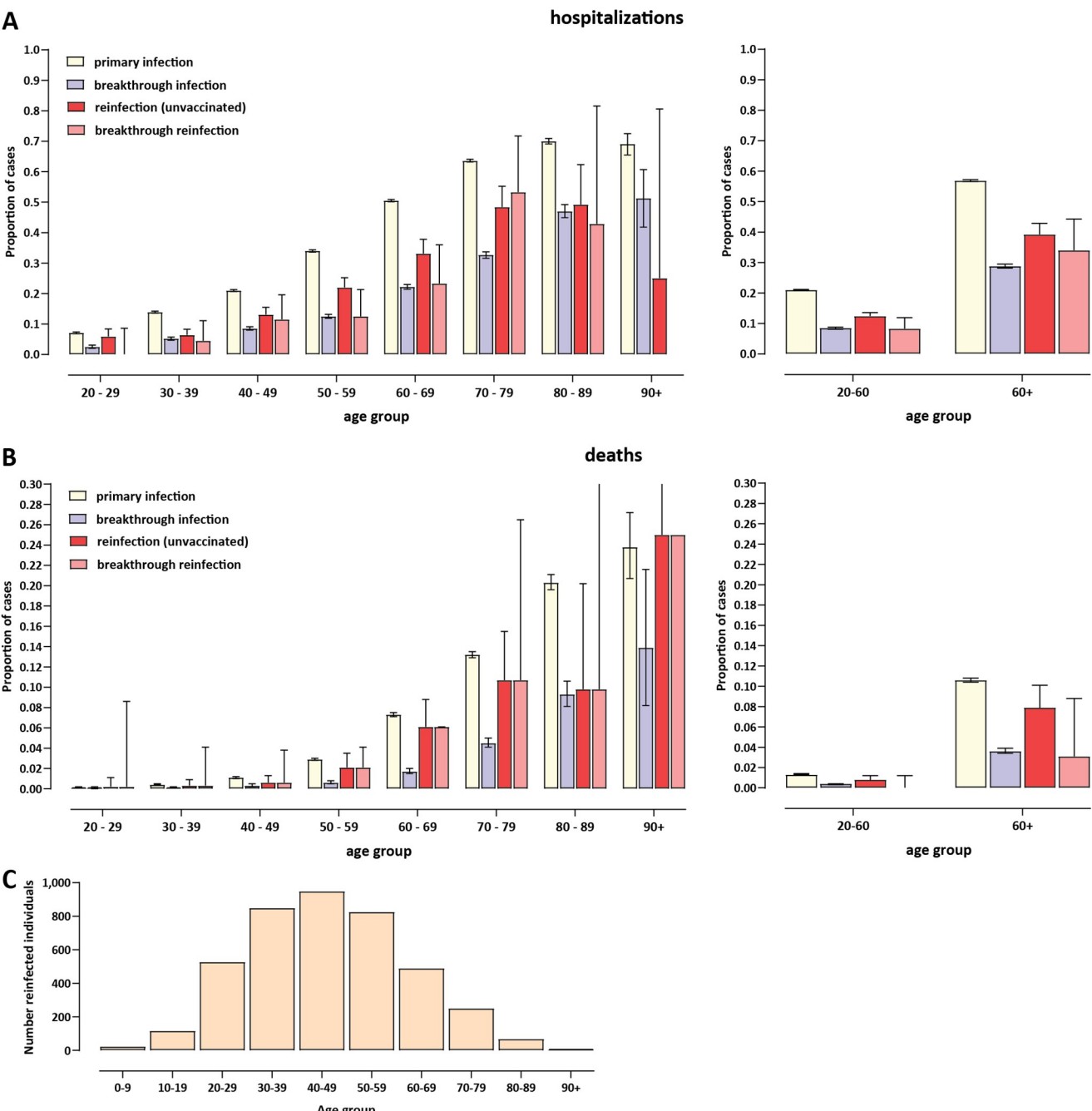

**Fig 3. Clinical severity of SARS-CoV-2 reinfections in previously infected individuals in Bulgaria.** (A) Percentage of hospitalizations among cases in primary infections, breakthrough infections (infections in vaccinated individuals), reinfections (divided into reinfections in the unvaccinated and breakthrough reinfections); (B) Percentage of deaths among cases in primary infections, breakthrough infections (infections in vaccinated individuals), reinfections (divided into reinfections in the unvaccinated and breakthrough reinfections). Binomial proportion confidence intervals were estimated using the Clopper-Pearson exact binomial interval method. (C) Age distribution of reinfected individuals.

## Acknowledgments

The authors would like to acknowledge the help of the Bulgarian Ministry of Health and Information Services for providing us with raw data about reinfections, demographics and vaccination status.

## Author Contributions

**Conceptualization:** Georgi K. Marinov, Antoni Rangachev.

**Data curation:** Georgi K. Marinov, Mladen Mladenov, Antoni Rangachev.

**Formal analysis:** Georgi K. Marinov, Mladen Mladenov.

**Investigation:** Georgi K. Marinov.

**Methodology:** Georgi K. Marinov, Mladen Mladenov, Antoni Rangachev.

**Project administration:** Georgi K. Marinov, Antoni Rangachev, Ivailo Alexiev.

**Software:** Georgi K. Marinov, Mladen Mladenov.

**Supervision:** Georgi K. Marinov, Antoni Rangachev, Ivailo Alexiev.

**Visualization:** Georgi K. Marinov.

**Writing – original draft:** Georgi K. Marinov, Mladen Mladenov.

**Writing – review & editing:** Georgi K. Marinov, Mladen Mladenov, Antoni Rangachev, Ivailo Alexiev.

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
