## [Decision Letter · Decision Letter 0]

27 May 2022

PONE-D-22-06223SARS-CoV-2 reinfections during the first three major COVID-19 waves in BulgariaPLOS ONE

Dear Dr. Marinov,

Thank you for submitting your manuscript to PLOS ONE. After careful consideration, we feel that it has merit but does not fully meet PLOS ONE’s publication criteria as it currently stands. Therefore, we invite you to submit a revised version of the manuscript that addresses the points raised during the review process.

Both the reviewer have recommended for revision of the manuscript and addition of some data. They also recommended for improvement of the english language.Please ensure that your decision is justified on PLOS ONE’s publication criteria and not, for example, on novelty or perceived impact.

We look forward to receiving your revised manuscript.

Kind regards,

Debdutta Bhattacharya

Academic Editor

PLOS ONE

Journal Requirements:

Reviewers' comments:

Reviewer's Responses to Questions

**Comments to the Author**

1. Is the manuscript technically sound, and do the data support the conclusions?

Reviewer #1: Partly

Reviewer #2: Yes

2. Has the statistical analysis been performed appropriately and rigorously? 

Reviewer #1: No

Reviewer #2: Yes

3. Have the authors made all data underlying the findings in their manuscript fully available?

Reviewer #1: No

Reviewer #2: Yes

4. Is the manuscript presented in an intelligible fashion and written in standard English?

Reviewer #1: Yes

Reviewer #2: Yes

5. Review Comments to the Author

Reviewer #1: Dear Authors

The manuscript was well written by the authors with minimum typographical errors. Authors should correct before submitting the revised manuscript by hiring a copyeditor.

Even the results shown less significant. Authors should include the co-morbidity status of the patients used in this study.

The demography of the samples collected may highlighted in detail.

The sample volume may be increased to get a very good model of prediction.

These suggestions may be addressed before final decision of the editor.

Reviewer #2: As the title suggest the study includes data from three major waves, it would be better to include the period of three waves individually in the manuscript to give a better picture.

The data collected is well analyzed statistically.

The literature has been well reviewed and included in the manuscript.

However, the manuscript needs revision for language and grammar to match the publication quality.

6. PLOS authors have the option to publish the peer review history of their article (what does this mean?). If published, this will include your full peer review and any attached files.

Reviewer #1: No

Reviewer #2: No

---

## [Author Response · Author response to Decision Letter 0]

14 Jun 2022

The response to reviewers is provided as a separate pdf file

---

## [Decision Letter · Decision Letter 1]

30 Aug 2022

SARS-CoV-2 reinfections during the first three major COVID-19 waves in Bulgaria

PONE-D-22-06223R1

Dear Dr. Marinov,

We’re pleased to inform you that your manuscript has been judged scientifically suitable for publication and will be formally accepted for publication once it meets all outstanding technical requirements.

Kind regards,

Debdutta Bhattacharya

Academic Editor

PLOS ONE

Additional Editor Comments (optional):

Reviewers' comments:

Reviewer's Responses to Questions

**Comments to the Author**

1. If the authors have adequately addressed your comments raised in a previous round of review and you feel that this manuscript is now acceptable for publication, you may indicate that here to bypass the “Comments to the Author” section, enter your conflict of interest statement in the “Confidential to Editor” section, and submit your "Accept" recommendation.

Reviewer #1: All comments have been addressed

Reviewer #2: All comments have been addressed

2. Is the manuscript technically sound, and do the data support the conclusions?

Reviewer #1: Yes

Reviewer #2: Yes

3. Has the statistical analysis been performed appropriately and rigorously? 

Reviewer #1: No

Reviewer #2: Yes

4. Have the authors made all data underlying the findings in their manuscript fully available?

Reviewer #1: Yes

Reviewer #2: Yes

5. Is the manuscript presented in an intelligible fashion and written in standard English?

Reviewer #1: Yes

Reviewer #2: Yes

6. Review Comments to the Author

Reviewer #1: Dear Authors

The comorbidity of patients were not given for the data shown in the manuscript. This may reflect the reinfection status of the patients. If possible kindly include the data in the article.

Reviewer #2: Manuscript is revised as per the comments. The comments have been well adressed. Revision for language and grammar to meet the publication quality has also been done.

7. PLOS authors have the option to publish the peer review history of their article (what does this mean?). If published, this will include your full peer review and any attached files.

Reviewer #1: No

Reviewer #2: No

---

## [Editor Report · Acceptance letter]

1 Sep 2022

PONE-D-22-06223R1 

SARS-CoV-2 reinfections during the first three major COVID-19 waves in Bulgaria 

Dear Dr. Marinov:

I'm pleased to inform you that your manuscript has been deemed suitable for publication in PLOS ONE. Congratulations! Your manuscript is now with our production department. 

Kind regards, 

on behalf of

Dr. Debdutta Bhattacharya 

Academic Editor

PLOS ONE